# Exploring the Interaction between Fractional Exhaled Nitric Oxide and Biologic Treatment in Severe Asthma: A Systematic Review

**DOI:** 10.3390/antiox12020400

**Published:** 2023-02-07

**Authors:** Tommaso Pianigiani, Lorenzo Alderighi, Martina Meocci, Maddalena Messina, Beatrice Perea, Simona Luzzi, Laura Bergantini, Miriana D’Alessandro, Rosa Metella Refini, Elena Bargagli, Paolo Cameli

**Affiliations:** Respiratory Diseases Unit, Department of Medicine, Surgery and Neurosciences, University of Siena, 53100 Siena, Italy

**Keywords:** nitric oxide, biomarker, severe asthma, biologic

## Abstract

Background: Fractional exhaled nitric oxide (FeNO) is a biomarker of airway inflammation associated with airway hyper-responsiveness and type-2 inflammation. Its role in the management of severe asthmatic patients undergoing biologic treatment, as well as FeNO dynamics during biologic treatment, is largely unexplored. Purpose: The aim was to evaluate published data contributing to the following areas: (1) FeNO as a predictive biomarker of response to biologic treatment; (2) the influence of biologic treatment in FeNO values; (3) FeNO as a biomarker for the prediction of exacerbations in patients treated with biologics. Methods: The systematic search was conducted on the Medline database through the Pubmed search engine, including all studies from 2009 to the present. Results: Higher baseline values of FeNO are associated with better clinical control in patients treated with omalizumab, dupilumab, and tezepelumab. FeNO dynamics during biologic treatment highlights a clear reduction in FeNO values in patients treated with anti-IL4/13 and anti-IL13, as well as in patients treated with tezepelumab. During the treatment, FeNO may help to predict clinical worsening and to differentiate eosinophilic from non-eosinophilic exacerbations. Conclusions: Higher baseline FeNO levels appear to be associated with a greater benefit in terms of clinical control and reduction of exacerbation rate, while FeNO dynamics during biologic treatment remains a largely unexplored issue since few studies have investigated it as a primary outcome. FeNO remains detectable during biologic treatment, but its potential utility as a biomarker of clinical control is still unclear and represents an interesting research area to be developed.

## 1. Introduction

Asthma is a chronic inflammatory disease affecting airways, known to affect about 300 million individuals worldwide [1]. Most asthmatic patients achieve control of disease through inhaled corticosteroids (ICS), combined or not with long-acting beta2-agonists bronchodilators (LABA), but 5–10% of them suffer from severe asthma [2]. Severe asthma is defined as asthma which requires treatment with high dose ICS, plus a second controller (and/or systemic corticosteroids) to prevent it from becoming ‘uncontrolled’ or which remains ‘uncontrolled’ despite this therapy [2].

Severe and poorly controlled forms are associated with higher morbidity and mortality [3].

Once clinical and functional diagnosis of asthma has been established, further evaluations, including a full allergological workup, should be done in order to identify possible specific triggers or predisposing conditions that may have an impact on asthma management. Moreover, the identification of a specific phenoendotype of disease could be crucial for therapeutic choices, especially in a severe asthma setting. Specific biomarkers, including peripheral blood eosinophils and fractional exhaled nitric oxide (FeNO) may help identify patients who are more likely to respond to specific biologics, allowing for a more personalized approach to the treatment of asthma. In this context, FeNO assessment has to be taken into consideration. The majority of patients with asthma, even in milder stages of the disease, show higher concentrations of FeNO than healthy controls, in correlation with high expression levels of the inducible NO synthase in airway epithelium, which appeared to be mainly associated with airway hyper-responsiveness and type-2 inflammation. Thus, the concentration of FeNO is regarded as a reproducible and non-invasive biomarker of airway inflammation [4].

The majority of clinical evidence supporting the use of FeNO can be categorized into the following areas: (1) diagnosis of asthma, (2) steroid responsiveness and dosing of inhaled corticosteroids (ICS), (3) monitoring asthma control, (4) medication adherence, and (5) choice of biologics for the treatment of severe asthma [5].

Nevertheless, the impact of FeNO in the management and prediction of exacerbations in patients treated with biologics, as well as the influence of different biologic drugs in the FeNO kinetics and its potential role for the prediction of exacerbations in patients treated with biologics, are still unclear.

The aim of our systematic review was to evaluate and collect published data investigating the accuracy of FeNO measurement in patients with severe asthma, in order to: (1) identify patients who are more likely to respond to a specific biologic; (2) evaluate the effect of biologic treatment in FeNO values; and (3) predict the exacerbations and hospitalizations in patients treated with biologics.

## 2. Data Collection and Analysis

The flow diagram of study selection and final inclusion in this systematic review, as well as the overview of type of studies included, is shown in Figure 1 and Table 1, respectively. The systematic search was conducted on the Medline database through the Pubmed search engine. We included all clinical studies from 2009 to the present, and we impose a restriction on language of publication accepting only English texts. For the study selection, we used the following keywords: “Feno and severe asthma; feno and biological treatment; feno and omalizumab and severe asthma; feno and mepolizumab and severe asthma, feno and Benralizumab and severe asthma, feno and dupilumab and severe asthma, feno and tezepelumab and severe asthma, feno and tralokinumab and severe asthma, feno and Lebrikizumab and severe asthma; severe asthma and predictor response and biological treatment and feno”.

We checked reference lists of all primary studies and review articles for additional references.

We divided the studies into three groups based on their outcomes:FeNO as a predictor of a good response to a specific biologic drug.FeNO modifications in severe asthmatic patients treated with biologic drugFeNO as a predictor of exacerbations and/or biomarker of asthma control in patients treated with biologic drugs.

We included randomized controlled trials (RCTs), observational retrospective and observational prospective study, both multicentric and monocentric, focused on adults affected with severe asthma. We included only studies with available full text.

We excluded case-reports, review, and pre-print studies.

We used a data collection form for study characteristics and outcome data, and we extracted the following study characteristics: study title, first name author, study design, journal and year of publication, primary and secondary outcomes of the study, biological treatment.

This systematic review has been registered with PROSPERO regeneration number 387651.

## 3. Findings

In total, 68 studies (12 observational prospective studies, 21 observational retrospective studies, 18 controlled trials, 4 post hoc analysis, 3 meta-analysis) were selected for preliminary evaluation: as shown in Figure 1, we excluded 10 studies, as the study design was not suitable for the objective of the present study. Therefore, 58 studies were finally included in our review. The complete list of articles included is reported in Table 2.

### 3.1. FeNO as a Predictor of a Good Response

#### 3.1.1. Omalizumab

Considering that anti-IgE omalizumab has been the first biologic drug to be approved for severe asthma, many studies investigated the potential role of FeNO in predicting a good clinical response to treatment. In spite of some conflicting reports, the majority of studies showed that higher values of bronchial FeNO may be associated to a better outcome in terms of clinical control and reduction of exacerbation rate. In the EXTRA study, the evaluation of prediction value of different biomarkers demonstrated that high levels of FeNO (≥19.5 ppb), blood eosinophils (≥260 cells/µL), and serum periostin (≥50 ng/mL) were associated with a greater treatment effect of omalizumab on exacerbation frequency [6]. Accordingly, Frix et al. reported that high baseline FeNO levels in patients with omalizumab therapy were associated with a better clinical control of asthma, expressed by the reduction of the Asthma Control Questionnaire (ACQ) after one year of treatment. Nevertheless, ROC analysis showed a modest accuracy of FeNO as a prediction factor for a significant ACQ change during the treatment (AUC 0.57, *p* > 0.05) [7]. In an elegant real-world study by Kavati et al., patients with severe allergic asthma were stratified according to each biomarker specific cut-off value: high (>300 cells/mL) and low (<300 cells/mL) blood eosinophils and intermediate-high (>25 ppb) and low (<25 ppb) FeNO. Although omalizumab was associated with a significant improvement in asthma control across pretreatment eosinophilic cell count and FeNO levels, the most significant improvement of the Asthma Control Test (ACT) score was observed at 12 months in both the high eosinophils and the intermediate-high FeNO subgroups [8]. Interestingly, another manuscript supported the implementation of FeNO analysis in the clinical practice since it demonstrated a fair cost-effectiveness for the identification of good responders prior to initiating a trial of omalizumab [9]. Other evidences supported the role of this biomarker as a predictive value of good response to omalizumab, showing that patients with higher baseline FeNO values reported most benefits overall, not only in terms of exacerbation risk and asthma control, but also regarding hospitalization rate, lung function, quality of life, and oral steroids use [10,11,12,13].

Finally, a very recent meta-analysis including 41 studies enclosed all the principal biomarkers (age, IgE, FeNO, and blood eosinophils) as potential predictors of omalizumab effectiveness. Moreover, though less significantly, the analysis showed that higher IgE and blood eosinophils, also resulted in higher levels of FeNO [14].

However, there were also some reports showing conflicting results. The large multicentric PROSPERO study showed that patients appeared to respond equally to omalizumab regardless of their baseline blood eosinophil or FeNO levels in terms of exacerbations rate, hospitalizations, lung function, and ACT scores [15]. In the same way, a multicentric prospective open-label study published by Hoch et al. and purposed to validate the Seasonal Asthma Exacerbation Predictive Index in children did not report any difference in FeNO levels between those experiencing an exacerbation or not during the fall season [16], as well as a retrospective report by Kallieri et al. in which elevated FeNO values were not correlated with a better response to treatment in terms of clinical control of disease.

Overall, most of the studies show that patients in therapy with omalizumab with high baseline FeNO levels have a better improvement in asthma control. However, in the PROSPERO study [16], as well as in retrospective studies [9,11], patients responded equally to omalizumab therapy regardless of their baseline FeNO levels.

These differences of results can be explained by many factors; first, the variability of the study populations among the studies and the different FeNO values identified as cut-off values for high and low FeNO patients. The studies by Casale, Mansur and Kavati chose a FeNO cut-off value of 25 ppb that was lower than Frix et al. (35 ppb), and in which high baseline FeNO levels correlated to better asthma control. Moreover, the EXTRA study adopted the FeNO baseline levels standardized by ATS guidelines with a cut-off of 19.5 ppb, showing a reduction of exacerbations in patients with higher baseline FeNO levels. Therefore, future studies will hopefully implement the approved cut off values of FeNO to standardize the subgroup analyses of severe asthmatic patients.

Another factor which can influence FeNO levels is systemic and inhaled steroids. In the study by Mansur et al., there was a very high proportion of patients treated with maintenance OCS (82%) in respect to other similar studies included in the review; this aspect can further explain the discrepancy of findings among the studies available in the literature.

Moreover, another study by Solidoro et al. investigated the value of FeNO as a predictive biomarker not only for asthma control, but also for airway obstruction reversibility in patients treated with omalizumab, proposing a cut-off value of 30.5 ppb [12]. It is feasible that different cut-off values of FeNO could provide a higher accuracy for different clinical outcomes.

In conclusion, FeNO measurement is a cost-effective predictor of omalizumab treatment response, which can be used to identify omalizumab responders prior to initiating a trial of omalizumab therapy, still considering the limits due to the difficulty of establish a standardized cut-off and the many confounding factors that can contribute to modifying FeNO concentrations.

#### 3.1.2. Mepolizumab/Benralizumab/Reslizumab

Anti-IL5 (mepolizumab and reslizumab) and anti-IL-5 receptor α (benralizumab) has proved to effective in severe eosinophilic asthma (SEA), characterized by serum eosinophils >300 cell/mm^3^ and/or sputum eosinophils >2–3%. Few studies are available concerning the potential role of FeNO as a predictor of good response to anti-IL5 drugs, because serum eosinophils appeared overall to be the best bioindicator for this purpose, and also the best in terms of cost-effectiveness.

Concerning data from RCTs, in the DREAM study, baseline FENO proved to be less closely associated with a response to mepolizumab than blood eosinophil count [17], and similar results were reported for benralizumab in the phase 2b dose-ranging study [18]; thus, FeNO evaluation was not included in the following phase 3 studies, which led to the approval of mepolizumab and benralizumab for the treatment of SEA [19].

Accordingly, the MEX trial, whose principal aim was to investigate the inflammatory profile of asthma exacerbations in mepolizumab-treated patients, did not report any significant differences of baseline FeNO values between patients with or without exacerbations during the time of observation [20].

In the same way, the preclinical studies and phase 2–3 RCTs of reslizumab did not report any data concerning the potential predictive role of FeNO in SEA patients.

Concerning evidence coming from real-world studies, in a multicenter retrospective cluster analysis, Yamada et al. identified five distinct phenotypes of severe eosinophilic asthma according to the variable FEV1 responsiveness to benralizumab. Patients with higher baseline FeNO values appeared to have a significantly better response in terms of FEV1 increase, but not in terms of clinical control of the disease, evaluated through ACT scores [21]. Nevertheless, other monocentric real world studies have reported that the clinical effectiveness of mepolizumab and benralizumab was independent of the baseline FeNO level [22], or that FeNO values > 40 ppb were even predictive for the identification of good responders to benralizumab treatment [23]. However, neither study performed a direct comparison of accuracy between blood eosinophils and FeNO.

On the basis of these findings, FeNO appeared not to provide an additional benefit in predicting response to anti-IL5 drugs in patients with SEA in respect to blood eosinophilic cells count. FeNO values are strictly related to the IL-4 and -13 pathways and much less to IL-5 expression: for this reason, even if FeNO remains one of the most affordable prognostic and severity biomarker in SEA, its reliability in predicting the response to anti-IL5 drugs may be reduced.

#### 3.1.3. Dupilumab

Dupilumab is the only biological treatment approved for severe asthma for which baseline FeNO values higher than 25 ppb and blood eosinophils > 150 cell/mm^3^ are required for prescription.

This evidence has been driven by the results reported by the phase 3 LIBERTY ASTHMA QUEST study, in which the greatest treatment benefits in terms of annualized rate of severe asthma exacerbations were observed in patients with increased baseline levels of blood eosinophils and FeNO > 25 ppb [24]. These findings were also confirmed by a post hoc analysis specifically focused on the identification of the best responders to dupilumab treatment [25]. Moreover, the steroid-sparing effect of dupilumab also appeared to be more pronounced in patients with increased FeNO values, though not in a statistically significant way [26].

The role of FeNO as a predictor of responsiveness of dupilumab treatment was further confirmed by post hoc analyses, a metanalysis and real-world studies, that also included the evaluation of other efficacy outcomes, such as pulmonary function tests [27,28,29,30].

Only one monocentric open-label study did not report any difference in terms of clinical effectiveness and steroid sparing effect between patients with FeNO higher or lower than 25 ppb: however, beyond the design, the study was also limited by the little sample size of the population (18 patients), which also includes a relevant quote of patients with chronic rhinosinusitis with nasal polyposis, which may influence the clinical response to dupilumab [31].

Overall, solid evidence supports the role of FeNO in predicting a good response to dupilumab in severe asthmatic patients: these results are mainly explained by the dual inhibition of the IL-4 and -13 pathways, which represents the most important source of NO production in respiratory airways. Therefore, FeNO can be considered a key tool to identify patients with severe type2-asthma who will best benefit from dupilumab treatment. More evidence is needed to evaluate whether other cut-off values of FeNO may be more specific for predicting the improvement of outcomes other than clinical control and exacerbation rate, such as a steroid-sparing effect, improvement of pulmonary function tests, bronchial hyperreactivity, and quality of life.

#### 3.1.4. Tralokinumab/Lebrikizumab

Anti-IL13 drugs such as tralokinumab and lebrikimuzab have been investigated as potential add-on treatments in severe asthma. In the phase 3 RCTs STRATOS 1 and 2, it was reported that baseline FENO 37 ppb or higher was the best biomarker to predict an enhanced response to tralokinumab in terms of severe exacerbation rate. These finding were considered clinically meaningful in STRATOS 1, but were not confirmed by STRATOS 2, leading substantially to the disapproval of tralokinumab for clinical practice [32].

Concerning lebrikizumab, Corren et al. showed that higher baseline FeNO values, as well as increased serum periostin concentrations, were also associated with a greater efficacy of lebrikizumab in improving FEV1 and in reducing the rate of severe exacerbations among patients receiving lebrikizumab than those receiving placebo. However, a greater intrapatient variability in baseline FeNO than in periostin levels was observed during the run-in period (mean coefficient of variation, 19.8% vs. 5.0%) [33].

Interestingly, in severe asthma patients treated with anti-IL13 drugs, elevated FeNO levels appeared to be related to a better clinical control. Despite the fact those drugs failed to obtain approval for clinical use, these findings underscore the correlation between FeNO values and IL-13 related airway inflammation, supporting the utility of FeNO in the choice of biologic treatment.

#### 3.1.5. Tezepelumab

Tezepelumab targets thymic stromal lymphopoietin (TSLP), a key alarmin expressed by epithelial cells of the respiratory tract in response to irritating and/or pro-inflammatory stimuli. It has been recently approved by the FDA for the treatment of severe asthma, and it is the first biologic drug that has potentially no prescriptive restrictions due to different asthma endotypes and/or biomarker expression.

In fact, tezepelumab was effective in improving asthma control and reducing the rate of disease exacerbations regardless the levels of T-2 biomarkers, including FeNO [34,35]. However, patients with higher values of FeNO, as well as those with higher blood eosinophils, appeared to show the best benefit from anti-TSLP treatment [36].

No real-world or observational studies are currently available on this topic in the literature.

To date, the correlation between FeNO values and TSLP is fully unexplored. However, when released by epithelial cells, TSLP acts as a trigger mainly, but not exclusively, for type 2 inflammation through multiple mechanisms (activation of dendritic cells with consequent differentiation of naive T cells to Th2 cells; activation and proliferation of type 2 innate lymphoid cells (ILC2); direct activation and degranulation of mast cells), leading to an overproduction of type2 cytokines, including IL-5, IL-4, and IL-13. Therefore, it is feasible to assume that a higher TSLP expression may be associated with an increased production of FeNO, even though the biomarker cannot be considered a reliable indicator for the prediction of tezepelumab response, considering the wide range of activity and target cells of TSLP.

### 3.2. FeNO Modifications in Severe Asthmatic Patients Treated with Biologic Drug

#### 3.2.1. Omalizumab

The only data available exploring the modification of FeNO during omalizumab treatment comes from observational cohort studies.

The majority of studies showed that omalizumab therapy led to a significant reduction of FeNO levels both in adults [11,13,37,38] and in children with severe allergic asthma [39]. These results were also confirmed by Frix et al., who reported a significant reduction of FeNO after just 16 weeks of exposure to omalizumab. Interestingly, the decrease of FeNO levels appeared to be progressive throughout the follow-up of five years, reaching a median reduction of 15.3 ppb [7].

Zietkowski et al., after 16 weeks of treatment with omalizumab, found a statistically significant decrease in FeNO that was also significatively correlated with the reduction of other T2 biomarkers, such as blood eosinophil cell count, serum ECP, and eotaxin. Such correlations were not observed in the group of patients not treated with omalizumab [40].

However, other studies failed to show significant variations of FeNO during omalizumab treatment. In a small study by Johansson et al., no differences were found after 16 weeks of treatment, even though a nearby significant reduction of FeNO was observed in the subgroup with allergen-driven hyperactivated basophils [41]. Ledford et al. showed that there were no statistically significant differences between treatment groups with omalizumab at weeks 12, 24, 36, and 52, as measured by the change from baseline in FeNO values [42].

Finally, only one study investigated the potential effect of omalizumab on alveolar nitric oxide concentration (CaNO), calculated through multiple-flow FeNO analysis, showing no significant variations after 16 weeks of treatment [43].

Overall, conflicting results are reported in the literature on this issue. Considering the differences in terms of FeNO measurement and the lack of studies primarily focused on investigating FeNO dynamics during omalizumab treatment, it appears that omalizumab use does not automatically lead to a reduction of FeNO levels. However, both in vitro and in vivo studies have demonstrated that omalizumab exposure led to a significant reduction of NO production, mainly through inhibition of the IL-4 pathway [44], while more conflicting results have been reported in IL-13 expression [45,46]. Interestingly, a study by Sellitto et al. investigated the impact of omalizumab on circulating T2-cytokines in non-asthmatic patients affected by chronic spontaneous urticaria. The results confirmed a significant reduction of IL-4 concentration, but not of IL-13, suggesting that IL-13 modifications during anti-IgE treatment may represent a secondary effect of the downregulation of other inflammatory pathways instead of a direct effect of the drug [47].

In conclusion, the overall influence of omalizumab on FeNO remains unclear, and future studies will have to take into account the potential effects of comorbidities and allergen exposure that may significantly influence FeNO levels.

#### 3.2.2. Mepolizumab/Benralizumab

Conflicting results have been published concerning the modifications of FeNO during treatment with anti-IL-5 agents. The majority of studies are focused on mepolizumab: no data are available from RCTs on this issue, except for the study by Haldar et al., that, however, explored the clinical and immunological effects of iv 750 mg mepolizumab and not of sc 100 mg formulation [48].

A large metanalysis, including 1457 patients from 13 studies, showed a significant reduction of FeNO levels in patients treated with mepolizumab [49], as well as other observational studies, some also with a large-sized population and with a multicenter design [50,51,52,53]. Interestingly, a real-world monocentric prospective study showed that the decrease in exhaled NO was characterized by a faster reduction of CaNO than bronchial NO, whose variation reached statistical significance only after 6 months of treatment [54].

On the contrary, many other reports showed no significant differences in FeNO values in mepolizumab-treated patients [55], even though the majority of these studies may be influenced by a small population size [56,57], retrospective design [58], a short follow-up [59], or comorbidities such as CRSwNP or bronchiectasis [60]. One study showed even a transient increase of FeNO values in patients treated with mepolizumab or benralizumab, but this increase was not associated to clinical deterioration [61].

Even though the evaluation of FeNO modifications was not one of its main outcomes, the MEX study showed that SEA patients treated with mepolizumab may show different FeNO behaviors that, moreover, were not even closely related to exacerbation risk, since their different inflammatory profile (high or low eosinophilic) were associated with higher or stable FeNO levels in respect to the baseline value [20].

Concerning benralizumab, only three studies demonstrated a significant reduction of FeNO levels after at least 6 months of treatment [22,62,63], while many other reports failed to show any significant differences [21,55,64,65,66]. However, the sample size of the study population was significantly larger in those showing a reduction of FeNO levels, not to mention the relevant heterogeneity of patients enrolled in the studies in terms of severity of disease and comorbidity. Interestingly, the study by Pelaia et al. also investigated the potential influence of atopic status on this issue, showing no differences between atopic and non-atopic patients in terms of FeNO values and modifications [63].

Only one study provided a comparison between mepolizumab and benralizumab, showing a much more pronounced reduction in mepolizumab-treated patients, but without any substantial clinical consequences [50].

Despite some conflicting reports, the sum of the published evidence generally showed a reduction trend of FeNO levels after mepolizumab or benralizumab treatment. Interestingly, the majority of studies are concordant in reporting a significant decrease of FeNO after at least four months of treatment, confirming that NO production is not primarily dependent on the IL-5 pathway. Unfortunately, no specific studies have been published regarding the impact of mepolizumab and benralizumab on the IL-4 and -13 axes.

#### 3.2.3. Dupilumab

Dupilumab systematically reduces FeNO levels in patients affected by severe asthma, and this reduction appeared to be quick and sustained throughout the treatment period. Castro et al. showed that patients who received dupilumab had greater reductions from baseline over the course of the intervention period in the FeNO and levels of total IgE (periostin, eotaxin-3, TARC) than patients who received a matched placebo [24].

In the study by Rabe et al., dupilumab treatment led to a suppression in the FeNO level by week 2, which was sustained during the 24-week intervention period. The percentage of patients with a FeNO level of less than 25 ppb increased from 44% at baseline to 84% at week 24 in the dupilumab group, whereas no meaningful change was observed in the placebo group (45% at both time points) [26]. Another multi-centre retrospective study showed that the treatment with dupilumab was associated to a significant improvement in FeNO value, which was evident already at 3 months of therapy and maintained after 12 months [31].

As expected, considering the mechanism of action of dupilumab, FeNO levels were systematically and steadily reduced during treatment, even though they appeared not to get zeroed as serum eosinophils after mepolizumab or, above all, benralizumab treatment. Importantly, the reduction of FeNO was described regardless of the coexistence of CRSwNP, atopic status, or OCS assumption. These findings are explained by the rapid, specific, and sustained downregulation of the IL-4 and IL-13 axes induced by dupilumab, which represents the main, but not unique, source of NO in respiratory airways [67,68].

#### 3.2.4. Tralokinumab/Lebrikizumab

In addition to dupilumab, anti-IL13 drugs also demonstrated a substantial reduction in FeNO in treated patients: in particular, tralokinumab appeared to reduce only FeNO concentration among the principal T2-biomarkers, such as blood and sputum eosinophilic cell count [69].

Accordingly, lebrikizumab also showed similar results, but the reduction was mainly pronounced in the high-periostin subgroup patients, which were also those who reported higher baseline FeNO values and that mostly benefited from treatment [33].

In the study of Corren et al., lebrikizumab produced a 19% mean decline in FeNO at week 12, as compared with a 10% increase with the placebo (*p* < 0.001). Among patients in the lebrikizumab group, there was a greater reduction in FeNO in the high-periostin subgroup than in the low-periostin subgroup (34.4% vs. 4.3%, *p* < 0.001 for the comparison of lebrikizumab with placebo in the high-periostin subgroup and *p* = 0.28 for the comparison in the low-periostin subgroup). The average FeNO value at baseline in the lebrikizumab group was 37 ± 3.8 ppb in the high-periostin subgroup and 25.3 ± 3 ppb in the low-periostin subgroup.

In general, data coming from anti-IL13 RCTs in severe asthma substantially confirmed the observations reported with dupilumab. Interestingly, the degree of FeNO reduction during lebrikizumab/tralokinumab therapy was lower than observed with dupilumab, supporting the evidence that a dual blockage of the IL-4/IL13 pathway provides a more extensive (and clinically efficient) downregulation of NO production than single IL-13 inhibition.

#### 3.2.5. Tezepelumab

In RCTs and following post hoc analyses, FeNO values were significantly decreased by tezepelumab treatment, regardless of inflammatory endotype, pulmonary functional assessment, steroid use, and the presence of CRSwNP [34,35,70]. Moreover, there were no notable differences in terms of speed or magnitude of the biomarker reductions between patients with or without CRSwNP or according to baseline FeNO, eosinophilic cell count and serum IgE, IL-5, IL-13, periostin, TARC, and TSLP. Nevertheless, the magnitude of reduction was maximal in those with eosinophils > 150 cell/mm^3^ and FeNO > 25 ppb. The decrease was evident after just four weeks after the first injection and was maintained during the observation period.

These findings further confirmed the broad anti-inflammatory activity of tezepelumab through the inhibition of TSLP; treated patients experienced a significant reduction of all biomarkers collected throughout the RCT, including FeNO, IgE, IL-5, and IL-13, demonstrating a direct activity of tezepelumab in downregulating the production of NO from the nasal and respiratory epithelium.

### 3.3. FeNO as a Predictor of Exacerbations and/or Biomarker of Asthma Control in Patients Treated with Biologic Drugs

#### 3.3.1. Omalizumab

Few studies are available describing the role of FeNO during omalizumab treatment. In a multicentric study investigating the long-term omalizumab effectiveness, Ledford et al. reported that in the group of subjects in which omalizumab was withdrawn, a rise in FeNO at week 12 was still predictive of an increased risk of exacerbation. However, in those patients who continued anti-IgE treatment, FeNO levels remained relatively stable and, in the case of exacerbation, the increase of FeNO, if present, was smaller in respect to patients withholding omalizumab [42].

According to this study, FeNO provides substantial help in the management of severe asthmatic patients during omalizumab treatment, since it appears to predict exacerbations with a fair-to-good accuracy. It is interesting to observe that patients stopping anti-IgE treatment showed an increase of FeNO only before or during an exacerbation, and not just after withholding omalizumab, suggesting that the reduction of exhaled NO values during the treatment are mainly related to an efficient downregulation of inflammatory pathways. On the other hand, a less pronounced increase of FeNO during omalizumab may still support the hypothesis of a treatment-related downregulation of NO production, even if partial.

#### 3.3.2. Mepolizumab/Benralizumab

Concerning FeNO dynamics during mepolizumab treatment, the MEX study showed that exacerbations may exert a different inflammatory profile which also influences FeNO levels; exacerbations characterized by a high eosinophilic burden (testified by an increase in sputum eosinophils percentage) were associated to a concomitant increase of FeNO. On the other hand, exacerbations sustained by non-predominantly eosinophilic inflammation (e.g., infection-driven) did not show significant variations of this biomarker [20]. This is the first and, so far, unique study to investigate the pathogenetic mechanisms underlying the exacerbations in severe asthmatic patients treated with biologics; thanks to its non-invasivity and reproducibility, FeNO measurement showed an interesting potential for the follow-up of SEA patients, since its variations may help not only to predict eosinophilic and/or T2-mediated exacerbations, but also to guide our therapy decisions in order to avoid unnecessary steroids or antibiotic prescriptions. However, more evidence is needed to confirm these findings, since the influence of anti-IL5 drugs in FeNO expression is still not clear. In fact, another paper has specifically investigated this topic, describing in a case series an increase of FeNO levels during mepolizumab and benralizumab. Despite the small sample size, the study provided interesting insights of FeNO dynamics during anti-IL5 treatments: in particular, the benralizumab subgroup showed a more relevant and faster increase of FeNO than mepolizumab, followed by an equally rapid decrease at 1 year of treatment. However, in this case, FeNO appeared not to be predictive for exacerbation, suggesting that IL-5 inhibition, especially through benralizumab, may lead to a provisional rebound of FeNO driven by an overexpression of the IL4/13 axes, with apparently no clinical consequences [61].

#### 3.3.3. Dupilumab

No studies are available for this topic.

#### 3.3.4. Tralokinumab/Lebrikizumab

No studies are available for this topic.

#### 3.3.5. Tezepelumab

No studies are available for this topic.

**Table 2 antioxidants-12-00400-t002:** List of articles included in the review.

Author	Type	Study Population	Target	Drug	Inclusion Criteria
Hanania et al., 2013 [6]	RCT	850	Allergic asthma	Omalizumab	Prediction of response
Frix et al., 2020 [7]	Observational retrospective	157	Allergic asthma	Omalizumab	Prediction of response; Variations during treatment
Kavati et al., 2019 [8]	Observational retrospective	473	Allergic asthma	Omalizumab	Prediction of response
Brooks et al., 2019 [9]	Observational prospective	NR	Allergic asthma	Omalizumab	Prediction of response
Solidoro et al., 2019 [10]	Observational retrospective	34	Allergic asthma	Omalizumab	Prediction of response; Variations during treatment
Mansur et al., 2017 [11]	Observational retrospective	45	Allergic asthma	Omalizumab	Prediction of response; Variations during treatment
Kurokawa et al., 2020 [12]	Observational prospective	16	Allergic asthma	Omalizumab	Prediction of response
Bhutani et al., 2017 [13]	Observational prospective	99	Allergic asthma	Omalizumab	Prediction of response; Variations during treatment
Y. Li et al., 2022 [14]	Meta-analysis	NR	Allergic asthma	Omalizumab	Prediction of response
Casale et al., 2019 [15]	Observational prospective	806	Allergic asthma	Omalizumab	Prediction of exacerbation during treatment
Hoch et al., 2017 [16]	RCT	486	Allergic asthma	Omalizumab	Prediction of response
Pavord et al., 2012 [17]	RCT	621	Eosinophilic asthma	Mepolizumab	Prediction of response
Castro et al., 2014 [18]	RCT	324	Eosinophilic asthma	Benralizumab	Prediction of response
McDowell et al., 2021 [20]	Observational prospective	145	Eosinophilic asthma	Mepolizumab	Prediction of response; Prediction of exacerbation during treatment
Yamada et al., 2021 [21]	Observational retrospective	64	Eosinophilic asthma	Benralizumab	Prediction of response; Variations during treatment
Hearn et al., 2021 [22]	Observational retrospective	229	Eosinophilic asthma	Mepolizumab Benralizumab	Prediction of response; Prediction of exacerbation during treatment
Watanabe et al., 2022 [23]	Observational retrospective	24	Severe type 2 asthma	Benralizumab	Prediction of response
Castro et al., 2018 [24]	RCT	1902	Uncontrolled moderate-to-severe asthma	Dupilumab	Prediction of response
Shrimanker et al., 2019 [25]	Post hoc analysis	606 + 1902	Eosinophilic asthma	Dupilumab Mepolizumab	Prediction of response
Rabe et al., 2018 [26]	RCT	210	Glucocorticoid dependent severe asthma	Dupilumab	Prediction of response; Variations during treatment
Pavord et al., 2020 [27]	Post hoc analysis	1037	Uncontrolled moderate-to-severe asthma	Dupilumab	Prediction of response; Variations during treatment
Carpagnano et al., 2022 [28]	Observational retrospective	12	Uncontrolled severe asthma	Dupilumab	Prediction of response; Variations during treatment
Yang et al., 2020 [29]	Meta-analysis	2992	Uncontrolled asthma	Dupilumab	Prediction of response
Rabe et al., 2022 [30]	Post hoc analysis	1902	Moderate-to-severe asthma	Dupilumab	Prediction of response
Campisi et al., 2021 [31]	Observational retrospective	18	Moderate-to-severe asthma	Dupilumab	Prediction of response; Variations during treatment
Panettieri et al., 2018 [32]	RCT	1140 + 770	Severe, uncontrolled asthma	Tralokinumab	Prediction of response; Variations during treatment
Corren et al., 2011 [33]	RCT	219	Severe, uncontrolled asthma	Lebrikizumab	Prediction of response; Variations during treatment
Corren et al., 2017 [34]	RCT	550	Severe, uncontrolled asthma with noneosinophilic inflammation	Tezepelumab	Variations during treatment
Menzies-Gow et al., 2021 [35]	RCT	1061	Severe, uncontrolled asthma	Tezepelumab	Prediction of response; Variations during treatment
Corren et al., 2022 [36]	RCT	550	Severe, uncontrolled asthma	Tezepelumab	Prediction of response; Variations during treatment
Cabrejos et al., 2020 [37]	Observational retrospective	345	Severe persistent allergic asthma	Omalizumab	Variations during treatment
Zietkowski et al., 2011 [38]	Clinical trial *	19	Severe persistent allergic asthma	Omalizumab	Variations during treatment
Silkoff et al., 2004 [39]	RCT	29	Allergic asthma	Omalizumab	Variations during treatment
Zietkowski et al., 2011 [40]	Clinical trial *	19	Severe persistent allergic asthma	Omalizumab	Variations during treatment
Johansson et al., 2018 [41]	Observational prospective	32	Allergic asthma	Omalizumab	Variations during treatment
Ledford et al., 2017 [42]	RCT	176	Moderate-to-severe asthma receiving omalizumab	Omalizumab	Variations during treatment; Prediction of exacerbation during treatment
Pasha et al., 2014 [43]	RCT	42	Uncontrolled moderate-to-severe asthma	Omalizumab	Variations during treatment
Haldar et al., 2009 [48]	RCT	61	Eosinophilic asthma	Mepolizumab	Variations during treatment
Li et al., 2021 [49]	Meta-analysis	1457	Eosinophilic asthma	Mepolizumab	Variations during treatment
Kayser et al., 2021 [50]	Observational retrospective	123	Eosinophilic asthma	Mepolizumab Benralizumab	Variations during treatment
Sposato et al., 2020 [51]	Observational retrospective	134	Eosinophilic asthma	Mepolizumab	Variations during treatment
Caminati et al., 2019 [52]	Observational retrospective	69	Eosinophilic asthma	Mepolizumab	Variations during treatment
Carpagnano et al., 2021 [53]	Observational retrospective	33	Severe eosinophilic allergic asthma	Mepolizumab	Variations during treatment
Cameli et al., 2020 [54]	Observational retrospective	27	Severe eosinophilic asthma	Mepolizumab	Variations during treatment
Izumo et al., 2020 [55]	Observational prospective	26	Severe asthma	Benralizumab	Variations during treatment
Farah et al., 2019 [56]	Observational prospective	20	Severe eosinophilic asthma	Mepolizumab	Variations during treatment
Kobayashi et al., 2021 [57]	Observational prospective	20	Severe eosinophilic asthma	Mepolizumab	Variations during treatment
Ramonell et al., 2021 [58]	Observational retrospective	47	Adult-onset severe asthma	Mepolizumab	Variations during treatment
Kalinauskaite Zukauske et al., 2019 [59]	Observational prospective	9	Severe non-allergic eosinophilic asthma	Mepolizumab	Variations during treatment
Crimi et al., 2021 [60]	Observational retrospective	32	Bronchiectasis + severe eosinophilic asthma	Mepolizumab	Variations during treatment
Pelletier et al., 2022 [61]	Observational retrospective	13	Severe eosinophilic asthma	Benralizumab Mepolizumab	Variations during treatment; Prediction of exacerbation during treatment
Padilla Galo et al., 2020 [62]	Observational prospective	42	Refractory eosinophilic asthma	Benralizumab	Variations during treatment
Pelaia et al., 2021 [63]	Observational prospective	111	Severe eosinophilic asthma	Benralizumab	Variations during treatment
Matsuno et al., 2020 [64]	Observational retrospective	17	Severe eosinophilic asthma	Benralizumab	Variations during treatment
Numata et al., 2020 [65]	Observational retrospective	24	Severe eosinophilic asthma	Benralizumab	Variations during treatment
Bagnasco et al., 2020 [66]	Observational retrospective	59	Severe uncontrolled asthma	Benralizumab	Variations during treatment
Russell et al., 2018 [69]	RCT	224	Moderate-to-severe asthma	Tralokinumab	Variations during treatment
Emson et al., 2021 [70]	Post hoc analysis	550	Severe, uncontrolled asthma	Tezepelumab	Prediction of response; Variations during treatment

* randomization not performed. RCT: randomized controlled trial; NR: not reported.

## 4. Study Limitations

Our study has some limitations. First, as already depicted in the Results section, very few studies were specifically focused on FeNO-related outcomes; moreover, the majority of data come from observational and real-world studies, leading to an unavoidable higher risk of reporting bias or referral bias. Second, we observed a wide variability in the selection of normal values of FeNO among the studies included in the review; this discrepancy may be related to the high degree of uncertainty concerning interpretation of FeNO values. Although the ATS guidelines define high, intermediate, and low FeNO levels in adults as >50 ppb, 25 to 50 ppb, different scientific societies and expert panels’ have identified different FeNO levels as cut-off values. Third, although FeNO is a reliable biomarker of type 2 inflammation in asthma, it is intrinsically affected by various factors such as gender, height, tobacco smoking and allergic sensitization, as well as ongoing treatment, which may have not been properly assessed in non-controlled studies. On the other hand, the design of the review was specifically focused on severe asthmatic patients treated with biologics, allowing us to include in the data analysis and discussion only patients with moderate-to high ICS dosage, therefore reducing the risk of bias related to inhaled treatments. The same assumption cannot be made for OCS-treated patients, since no study has specifically investigated the potential variation of FeNO due to systemic steroid treatment.

## 5. Conclusive Remarks

The overall role of FeNO in the management of severe asthmatic patients undergoing biologic treatment is still unclear. The majority of studies is focused on the utility of this biomarker in predicting the response to treatment, showing, beyond some conflicting results, that higher baseline values of FeNO are generally associated to a greater benefit in terms of reduction of exacerbation rate and improvement of clinical control, especially for omalizumab, dupilumab, and tezepelumab. However, no clear findings are available concerning the predictive value of FeNO for respiratory functional or steroid-sparing effects, as well as for quality of life outcomes (Figure 2).

During biologic treatment, the FeNO dynamics and interpretation is still a matter of debate because very few studies have investigated this specific issue. Interestingly, during anti-IL5 treatment, unlike blood eosinophils that are markedly reduced and sputum eosinophils that are undetectable, FeNO remains detectable even in patients treated with anti-IL4/13 and anti-IL13. However, the interpretation of FeNO values during biological treatment is still a largely unexplored issue, as well as its value in the context of drug shift, which will probably be one of the major research areas in the next few years.

Considering the non-invasivity, cost effectiveness, and reproducibility of this biomarker, future studies will have to address the potential of FeNO in the follow-up of severe asthmatic patients treated with biologics, in an optic of personalized and endotype-driven management.

## Figures and Tables

**Figure 1 antioxidants-12-00400-f001:**
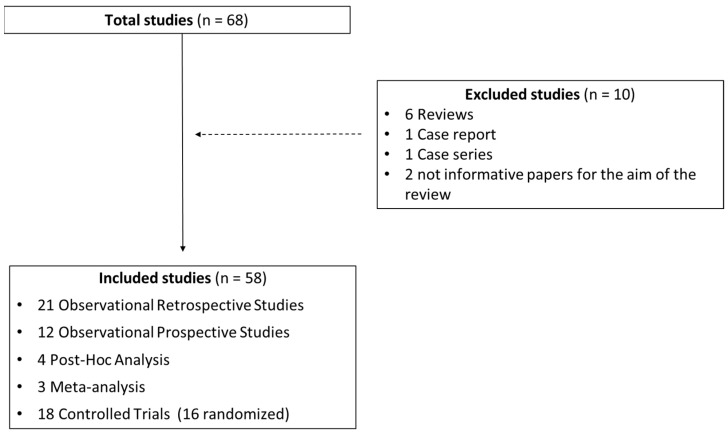
Flow diagram for study selection and inclusion for systematic review.

**Figure 2 antioxidants-12-00400-f002:**
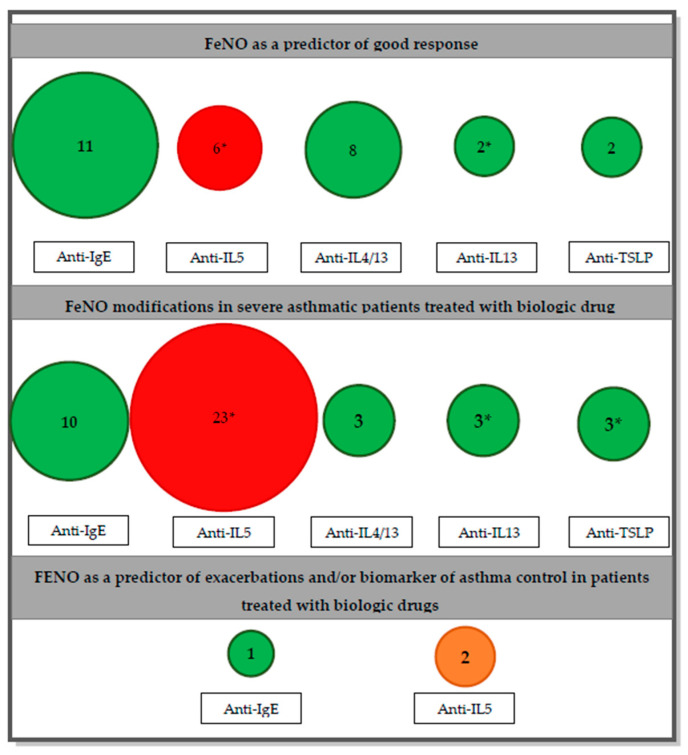
The size of the circles and the number inside reflect the number of articles reviewed per drug. The color reflects the agreement of the results with the points in examination (Green = concordance; Red = discordance; Orange = no diriment data). *: presence of RCT between the study reviewed.

**Table 1 antioxidants-12-00400-t001:** Overview of studies selected for systematic review, subdivided according to study design.

**Total Studies 68**	**Excluded Studies**	**10**	**Included Studies**	**58**
	Case report	1	Observational retrospective studies	21
	Reviews	6	Observational prospective studies	12
	Case series	1	*Post hoc* analysis	4
	Not informative papers for the aim of the review	2	Meta-analysis	3
			Controlled trials (16 randomized)	18

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
