# Peer review of "Exploring the Interaction between Fractional Exhaled Nitric Oxide and Biologic Treatment in Severe Asthma: A Systematic Review"

_antioxidants, 2023, doi:10.3390/antiox12020400_

Round 1

Reviewer 1 Report

Figure 1 should better changed into  a table.

The lack of studies  on FeNO measurements is in children due to the difficulty to perform this assay. Could the author discuss this possibility?

The review would benefit from a figure in which the different data are summarized. In fact as it is it is not easy to follow up the different results.

Moreover, in some studies only allergies without asthma were analyzed. The title refer to severe asthma.

Could the authory subdivide the studies in analysis of severe and not severe asthma with and without allergic component? Controls with a nd without allergies?

This would give a more structured analysis.

Author Response

Thank you for your suggestions to improve our paper. Below you can find the point by point answer to the questions:

  1. “Figure 1 should better changed into table”.

We thank you again for the advice. We're afraid that we have to maintain Figure 1 in the paper as a criteria for the definition of a systematic review. However, we have included another Table in the text as you suggested. 

Total Studies
= 68

Excluded Studies

10

Included Studies

58

case report

1

observational Retrospective Studies

21

reviews

6

observational Prospective Studies

12

case series

1

Post-Hoc Analysis

4

not informative papers for the aim of the review

2

Meta-Analysis

3

Controlled Trials (16 randomized)

18

  1. “The lack of studies on FeNO measurements is in children due to the difficulty to perform this assay. Could the author discuss this possibility?”

Thank you for the question. FeNO measurement can be useful for diagnosis and follow-up of asthma treatment in children, thanks to its non-invasivity and reproducibility. Nevertheless there is a lack of studies about this test in children in literature, especially in the topic of severe asthma. In our opinion, this lack could be due to the lack of standardization of FENO measurement in children. In addition, it should be considered that children do not always have good compliance in performing the examination, especially if they are very young, as you suggested. Moreover, very few studies have been conducted on this issue in patients < 18 years old: we agree with you concerning the potential utility to conduct a specific review (maybe a scoping review) on this topic.

However, as we have stated in the Data collection section, our review was focused on adult patients and, therefore, we believed that including data or studies concerning FeNO in children could be beyond the scope of this paper

  1. “The review would benefit from a figure in which the different data are summarized. In fact as it is it is not easy to follow up the different results.”

Thanks for the suggestion. We proceeded to add an image summarizing the results in our paper. You can find the figure in the attached document.

  1. "Moreover, in some studies only allergies without asthma were analyzed. The title refer to severe asthma."

Thank you for your observation. The paper included in the Review for data analysis were all focused on adult severe asthma, treated with biologic drugs: in the Findings section, as you correctly underscored, we have also cited data coming from studies conducted on CSU or CRSwNP as treatment indication but only to explain and discuss the findings reported in severe asthma studies. Therefore, these studies were not included in the data analysis, as you can check on Table 2

  1. "Could the authory subdivide the studies in analysis of severe and not severe asthma with and without allergic component? Controls with and without allergies? This would give a more structured analysis."

Thank you for your suggestion. As already stated in the text and depicted in Table 2, only studies focusing on adult severe asthma were included in the systematic review for data analysis. The studies concerning non severe-asthmatic patients (e.g., patients affected with CSU and treated with omalizumab or patients affected with CRSwNP and treated with omalizumab and/or dupilumab) were NOT included in the data analysis and were cited only to discuss and analyze the findings coming from severe asthma studies. Regarding the atopic status, we totally agree with you concerning the potential influence played by this factor on FeNO values: however, the specific inclusion and prescription criteria for the different biologic drug intrinsically addressed the selection of severe asthmatic patients with or without atopy, especially for omalizumab-treated patients. Concerning the other drugs, no specific data on FeNO values according the atopic status has been produced so far, but the majority of RCTs and multicentric observational study included the atopy as covariate, as already discussed in the text.

Reviewer 2 Report

In general, an excellent job and an excellent manuscript. Congratulations to the authors. However, there are a few points that should be clarified.

• The conclusion of the work presented in the abstract does not really match the conclusion in the main text. In my opinion, the conclusion in the main text is much more accurate.

• Limitations of work not provided. In my opinion, one of the most important (unfortunately unavoidable) shortcomings that arose due to the nature of this work, is the lack of information on the factors that can modify the concentration of FeNO (valid for all disused drugs). This needs to be mentioned as a limitation and discussed in the discussion section as well. The most important factors modifying the FeNO concentration are a dose of corticosteroids, time of the last allergen exposure, smoking, respiratory viruses, bacteria, and concomitant atopic diseases.

• The incorrect statement is in lines 261-267 because there is no direct evidence. Only assumptions are based on the indirect comparison. I would suggest that this paragraph be corrected.

Author Response

Thank you for your suggestions to improve our paper. Below you can find the point by point answer to the questions:

  1. “The conclusion of the work presented in the abstract does not really match the conclusion in the main text. In my opinion, the conclusion in the main text is much more accurate.”

Thank you for the suggestion. As you advised, we have modified the conclusion section of the Abstract as it follows:

“Conclusions: Higher baseline FeNO levels appear to be associated to a greater benefit in terms of clinical control and reduction of exacerbation rate, while FeNO dynamics during biologic treatment remains a largely unexplored issue as much as few studies have investigated it as a primary outcome. FeNO remains detectable during biologic treatment but its potential utility as biomarker of clinical control is still unclear and represents an interesting research area to be developed.”

  1. “Limitations of work not provided. In my opinion, one of the most important (unfortunately unavoidable) shortcomings that arose due to the nature of this work, is the lack of information on the factors that can modify the concentration of FeNO (valid for all disused drugs). This needs to be mentioned as a limitation and discussed in the discussion section as well. The most important factors modifying the FeNO concentration are a dose of corticosteroids, time of the last allergen exposure, smoking, respiratory viruses, bacteria, and concomitant atopic diseases.”

Thank you for your advice that will help us to improve our review. According to your suggestion, we have decided to include in the paper a subsection entitled “limitations”

"Limitations

Our study has some limitations. First of all, as already depicted in the Results' section, very few studies were specifically focused on FeNO-related outcomes; moreover the majority of data comes from observational and real-world studies, leading to an unavoidable higher risk of reporting bias or referral bias. Second, we observed a wide variability in the selection of normal values of FeNO among the studies included in the Review: this discrepancy may be related to the high degree of uncertainty concerning interpretation of FeNO Values. Although the ATS guidelines define high, intermediate, and low FeNO levels in adults as >50 ppb, 25 to 50 ppb, different scientific societies and expert panels' have identified different FeNO levels as cut-off values. Third, although FeNO is a reliable biomarker of type 2 inflammation in asthma, it is intrinsically affected by various factors such as gender, height, tobacco smoking and allergic sensitization, as well as ongoing treatment, that may have not been properly assessed in non-controlled studies. On the other hand, the design of the ewview was specifically focused on severe asthmatic patients treated with biologics, allowing us to include in the data analysis and discussion only patients with moderate-to high ICS dosage, therefore reducing the risk of bias relted to inhaled treatments.  The same assumption cannot be mde for OCS-treated patients, since no study has specifically investigated the potential variation of FeNO due to systemic steroid treatment."

  1. “The incorrect statement is in lines 261-267 because there is no direct evidence. Only assumptions are based on the indirect comparison. I would suggest that this paragraph be corrected.“

Thank you for your comment. We agree with your suggestion and we therefore corrected the paragraph as it follows:

“Interestingly, also in severe asthma patients treated with anti-IL13 drugs, elevated FENO levels appeared to be related to a better clinical control. Despite those drugs failed to reach the approval for clinical use, these findings underscore the correlation between FENO values and IL-13 related airway inflammation , supporting the utility of FeNO in the choice of biologic treatment”.

Reviewer 3 Report

The paper gives an interesting overview of studies on FeNO and biological treatment. I don't have any major comments. The manuscript reads well. 

Author Response

Thank you for taking the time and effort necessary to review the manuscript. We sincerely appreciate your feedback as well as all the suggestions, which helped us to improve the quality of the manuscript.